# Email Reminders Increase the Frequency That Pet Owners Update Their Microchip Information

**DOI:** 10.3390/ani8020020

**Published:** 2018-01-31

**Authors:** Katie Goodwin, Jacquie Rand, John Morton, Varun Uthappa, Rick Walduck

**Affiliations:** 1School of Veterinary Science, University of Queensland, Gatton, QLD 4343, Australia; j.rand@uq.edu.au (J.R.); john.morton@optusnet.com.au (J.M.); 2Australian Pet Welfare Foundation, Kenmore, QLD 4069, Australia; jacquie@petwelfare.org.au; 3Jemora Pty Ltd., Geelong, VIC 3220, Australia; 4Central Animal Records, Keysborough, VIC 3173, Australia; varun@microchips.com.au (V.U.); rick@car.com.au (R.W.)

**Keywords:** microchip, stray, euthanasia, age, socioeconomic index, breed, cat, dog, Australia, characteristics

## Abstract

**Simple Summary:**

Many stray animals presented to shelters in several first world nations have incorrect contact details associated with their microchip and, consequently, cannot be reunited with their owners. This study investigated whether sending email reminders increased the frequency that pet owners updated their contact details on an Australian microchip database, and characterized the cat and dog population on this database. Email reminders were found to be effective at increasing the frequency that pet owners updated their contact details; frequency of updates also varied according to species, pet age, state or territory and socioeconomic differences. The information gained from this study can be used to increase owner compliance in keeping pet microchip contact details up to date, and therefore increase reclaim percentages of stray animals.

**Abstract:**

Stray animals with incorrect microchip details are less likely to be reclaimed, and unclaimed strays are at increased risk of euthanasia. A retrospective cohort study was performed using 394,747 cats and 904,909 dogs registered with Australia’s largest microchip database to describe animal characteristics, determine whether annual email reminders increased the frequency that owners updated their information, and to compare frequencies of microchip information updates according to pet and owner characteristics. More than twice as many dogs (70%) than cats (30%) were registered on the database; the most numerous pure-breeds were Ragdoll cats and Staffordshire Bull Terrier dogs, and the number of registered animals per capita varied by Australian state or territory. Owners were more likely (*p* < 0.001) to update their details soon after they were sent a reminder email, compared to immediately before that email, and there were significant (*p* < 0.001) differences in the frequency of owner updates by state or territory of residence, animal species, animal age, and socioeconomic index of the owner’s postcode. This research demonstrates that email reminders increase the probability of owners updating their details on the microchip database, and this could reduce the percentages of stray animals that are unclaimed and subsequently euthanized.

## 1. Introduction

Australia has one of the highest rates of pet ownership worldwide, with an estimated 3.3 million cats and 4.2 million dogs belonging to 63% of Australian households [1]. In the United States (US), 44% of households own a cat or dog [2], and approximately 50% of United Kingdom (UK) households have a cat or dog [3]. Such high pet ownership rates in Western countries are associated with a substantial number of stray animals admitted to shelters and pounds each year. In 2015–2016, 45,256 dogs and 55,570 cats were presented to the Royal Society for the Prevention of Cruelty to Animals (RSPCA) in Australia [4], an estimated 6.5 million cats and dogs were received by the American Society for the Prevention of Cruelty to Animals (ASPCA) in the US [2], and the RSPCA UK received 129,602 cats and dogs [3]. Strays admitted via the general public and municipal contracts accounted for approximately 60% of all dog and cat admissions to RSPCA shelters in Australia [5,6], 26% of dogs and 42% of cats admitted to UK animal rescue organizations during 2010 [7], approximately 40–60% of animal intakes to shelters in the US [8], and 68% of cats and dogs admitted to shelters in Spain [9].

Stray pets that are not reunited with owners are at risk of euthanasia. In 2015–2016, 39% of dogs and 5% of cats admitted to the RSPCA Australia were successfully reunited with owners (‘reclaimed’), and 21% of unclaimed dogs and 30% of unclaimed cats were euthanized [4]. Nationally, 48% of all admitted dogs from Australian municipal councils, animal welfare organizations and rescue groups were reclaimed and 21% were euthanized (40% of unclaimed dogs) in the 2012–2013 financial year [10]. In ASPCA shelters, approximately 18% of stray dogs and 3% of stray cats were reclaimed while 23% of cats and dogs were euthanized in 2015–2016 [2]. In Spanish shelters, 17% of dogs and 4% of cats were reclaimed while 13% of dogs and 23% of cats were euthanized [9].

Currently, in all Australian states and territories except the Northern Territory (NT) and South Australia (SA), it is mandated that dogs and cats must be microchipped at the time of registration with local government (council) or transfer of ownership (Appendix A) [11,12,13,14,15,16,17,18,19,20]. It was estimated that 76% of Australian dogs and 64% of owned cats were microchipped in 2013 [1]. The act of implanting pet microchips is highly regulated, and animal and owner information and contact details must be provided to a licensed registry within a specified time following implantation. However, it is the responsibility of the owner to ensure pet and owner details (including owner contact details) remain up to date [15]. There are currently six licensed microchip databases in Australia that allow pet owners to access and update the details associated with their pet’s microchip: Central Animal Records, Australasian Animal Registry, Petsafe, New South Wales Pet Registry (compulsory for all New South Wales cats and dogs [14]), Home Safe ID and Global Micro Animal Registry [21,22,23,24,25,26].

Microchipping is a method of permanent identification that cannot be removed, dislodged or lost without surgical intervention, unlike collar identification tags and other tracking devices. Reclaim percentages for stray pets differ markedly between microchipped and unmicrochipped animals. In 2014, in RSPCA Queensland shelters, 80% of microchipped dogs versus 37% of unmicrochipped dogs, and 51% of microchipped cats versus 5% of unmicrochipped cats were reclaimed [27]. Although factors other than presence of a microchip could have contributed to these differences (for example, microchipping could be considered an indicator of responsible pet ownership), these results highlight the importance of microchips in reuniting owners and pets. Similarly, in a 2009 study of USA shelters, reclaim percentages for microchipped dogs were 2.5 times higher than for unmicrochipped animals [28], and a study of shelters in Spain showed dogs were three times more likely to be reclaimed if microchipped [9].

Nonetheless, 37% of stray animals presented to RSPCA Queensland during 2012–2013 [27], 21% of strays presented to 53 North American shelters during 2007–2008 [28], and 13% from the UK’s Dogs Trust during 2015–2016 [29] had incorrect owner contact information associated with their microchip. In addition, a survey in the UK found that 31% of all respondents whose contact details had changed had not updated their pets’ microchip details, despite knowing their details on the database were incorrect [30]. In a study of admissions to RSPCA Queensland, the odds of microchipped animals being reclaimed when microchip contact details were correct were 31 and 12 times higher for cats and dogs, respectively, than if contact details were incorrect. In turn, odds of being reclaimed for cats and dogs with incorrect details were 10 and 4 times higher, respectively, when compared to unmicrochipped animals [27].

Stray pets with incorrect microchip owner contact details also increase demands on shelter and municipal resources. Increased costs are incurred and resources are expended reuniting stray animals with owners because, when compared to pets with correct microchip details, the time taken to reclaim these pets is longer or reuniting is not possible at all [27]. Increased duration of stay in shelters requires more housing and husbandry resources and decreases space available for surrendered animals. In addition, more time is spent attempting to locate owners of stray pets with incorrect microchip owner contact details [27]. Knowledge of strategies that increase the accuracy of owner contact information on microchip databases would assist database companies in providing more accurate information to shelters and municipal animal facilities. This would facilitate more efficient allocation of resources in reuniting lost animals with their owners, and would reduce the risk of euthanasia of lost pets.

Thus, it is clear that correct contact details play a significant role in the proportion of stray animals reclaimed and euthanized, not just in Australia, but in several first world nations. Increasing the proportion of pets with correct contact details is positively correlated with being reclaimed and negatively correlated with euthanasia in shelters. However, strategies aiming to increase the accuracy of pet owners’ contact details on microchip databases have not been evaluated. Initiatives designed to increase microchip compliance include the US and UK’s National Microchipping Month every June [31,32], competitions for newly microchipped pets to win prizes sponsored by pharmaceutical companies and microchip manufacturers [33], and specific campaigns such as the RSPCA UK’s #chipneck to educate owners about their microchipping legal responsibilities [34]. However, these initiatives focus mainly on increasing microchip implantation compliance, rather than updating of contact details. There are fewer initiatives aimed at increasing updating of details, such as legislation mandating details are kept up to date, council enforcement notices and advertising by the British Veterinary Association in the UK [29], and National Check the Chip Day in the US [35]. Central Animal Records sends annual ‘anniversary’ emails on the anniversary of the pet’s registration on the database to remind owners to update their pets’ microchip details. However, it is unknown what effect these emails have on the frequency of updates [21]. In addition, to the authors’ knowledge, no studies assessing other potential determinants of updating microchip details have been published.

Central Animal Records (CAR) is Australia’s largest microchip database [21]. Therefore, identification of species, breed, sex and age characteristics of such a substantial proportion of the Australian cat and dog population may facilitate an understanding of how these animal characteristics differ from reported shelter demographics, especially breeds and age groups overrepresented in shelters [6,27,36]. For example, studies from several first world nations show that adult pets, mixed-bred animals, and purebred Staffordshire bull terriers are the most numerous categories in shelters [6,9,28,36].

The aims of this study were to describe the characteristics of cats and dogs registered on an Australian national microchip database, to investigate whether sending email reminders increased the frequency that owners updated their contact details on the microchip database, and to determine whether there were relationships between the frequency of updates and species (cat or dog), pet age, state/territory, and socioeconomic status of the owner’s locality.

## 2. Experimental Section

### 2.1. Study Overview

A retrospective cohort study was performed using cats and dogs that had a microchip implanted from 1 January 2008 to 31 December 2016 and which were then registered on the CAR database. Each animal was registered only once; once registered, annual re-registration was not required.

### 2.2. Selection of Study Population

All cats and dogs with a microchip implanted from 1 January 2008 to 31 December 2016 were selected, and their details were used as recorded in the database on 14 August 2017. Of the animals selected (434,355 cats, 1,066,810 dogs), further exclusions were made if the owner had >5 cats or dogs in the database, the owner’s address was not in Australia or the owner’s country of residence was uncertain, the animal shared the same microchip number with another animal, or the animal had more than one record with the same animal identification and microchip number. Animals whose owner had >5 cats or dogs were excluded to avoid including both animals registered by breeders or shelters and animals with CAR recovery records. Recovery records were created when a microchip number was searched for, in the case of lost or stolen pet. If not recorded on CAR (because the number had been previously recorded on one of the other Australian microchip registries), a record on CAR was created, indicating that the registry contained complete, current animal owner information. The 394,747 cats and 904,909 dogs not excluded by the above criteria were used for analysis of animal characteristics (Table 1 and Table 2).

Date of birth was recorded for 368,224 cats and 848,138 dogs. For 51 of these cats and 116 of these dogs, the recorded implantation date and/or the recorded registration date were before the animal’s date of birth, so these animals were excluded from analyses where age was used. A software update on CAR has subsequently been implemented to prevent this occurring.

### 2.3. Anniversary Email Reminders

From 2013, CAR sent annual emails reminding owners to update their and their pets’ details on each anniversary (day 365 or 366) after the date their pet was first registered on the CAR database, except for a pause from 31 March to 4 July 2016. This pause occurred due to a change in CAR’s information technology systems. Owners with no email address recorded at the time of their pet’s anniversary did not receive any anniversary reminders.

### 2.4. Definitions

The updating of owner details when emails were sent was compared to the period when emails were not sent (emails were not sent from 31 March 2016 to 4 July 2016). Updating frequency between pets whose owners had provided an email address was also compared to pets whose owners had not. An update of owner details was defined as having occurred whenever any of the owner’s details (i.e., name, phone, email address, home address, etc.) were updated. Due to privacy legislation regarding owner contact details, it was unknown what specific owner details were updated in each update event. Where an owner made multiple updates on the same date, this was treated as one update event.

There are several methods available for owners to update their pets’ microchip details with the CAR database, other than through email reminders. Owners without an email address listed can create login access via the CAR website, or details can be updated via phone and mail. Thus, owners without an email on file can readily update their details online, and non-internet users have an alternative available.

Owner email address status was assessed according to whether or not an email address was listed in the CAR database as at 14 August 2017. All text strings containing the ‘@’ symbol were classified as email addresses.

Purebred animals were defined as those with one breed listed on their record, and mixed-bred animals were defined as those animals whose owner had listed a primary and secondary breed on their pet’s record, or a mixed breed as the primary breed. For the purposes of this study, domestic short, medium and long hair cat breeds were considered mixed-bred. Socioeconomic statuses of owners’ localities were described using the most current (2011) Australian Bureau of Statistics Socioeconomic Indexes for Areas (SEIFA) [37]; the index of relative socioeconomic advantage and disadvantage for each owner’s postcode was used.

Owners commonly did not record instances of animal deaths, losses or thefts on the database. Accordingly, we could not account for many of these animals in analyses. Rather, we assumed all animals were alive and in the care of the owner (as recorded in the database at 14 August 2017) for at least 540 days from the animal’s registration date.

### 2.5. Data Collection

Details regarding species, microchip implant or registration date, breed, sex and neuter status, owner postcode and state, and data update date were extracted from the CAR database. All updates of owner details from 1 March 2015 to 17 August 2017 were extracted (including updates during the period from 31 March 2016 to 4 July 2016, when email reminders were not sent; Table 2). Updates preceding this date were not included because a different CAR information technology system had been used prior to 1 March 2015, affecting what data could be accessed. Events of owners updating pet details (for example age, breed, color, change in neuter status, deceased or alive) were not used. Animal characteristics (age, sex, current neuter status, breed) and owner characteristics (owner’s state and postcode, and whether or not an email address was recorded) were extracted with details as recorded at 14 August 2017.

### 2.6. Analysis of Owner Details’ Updates

To study owner details’ update patterns from registration date, only animals registered on the CAR database from 1 March 2015 to 24 February 2016 were included (Table 2). The first of March 2015 was selected because registration dates (and hence dates that reminder emails were sent) were only available for animals registered from 1 March 2015. We chose 24 February 2016 to ensure that updates of owner details were available for at least 18 months from each animal’s registration date (as there are 540 days from 24 February 2016 to 17 August 2017). For animals registered after 24 February 2016, updates of owner details were not available for at least 540 days, so these animals were not used for these analyses.

### 2.7. Statistical Analyses

To determine whether anniversary emails were associated with updating of owner details, we had three sources of evidence. Firstly, if anniversary emails were effective, an increase in the numbers of updates each day would be expected around 12 months after the animals’ dates of registration, and around 12 months after their previous anniversaries. Secondly, such increases would not be expected for animals whose owners had no email address listed on the CAR database. Thirdly, no such increase would be expected for animals whose owners had an email address listed but whose anniversary fell in the period from 31 March to 4 July 2016, when anniversary emails were not sent.

Generalized Estimation Equations (GEE) were used to assess associations between potential determinants of an update of owner details. For analyses of such updates from date of registration on the CAR database, the unit of analysis was the animal-month, where each animal provided 18 months, one each from days 2 to 30, 31 to 60, 61 to 90, and so on to 511 to 540 days after date of entry. GEE models were fitted using the -xtgee- command in Stata (version 15, StataCorp, College Station, TX, USA). The individual animal was the panel variable, with animal-months within each animal treated as repeated measures. Each animal-month was categorized as having either at least one update of owner details or none; this was the dependent variable. The binomial error distribution and logit link were fitted, and the exponentiated coefficients were interpreted as odds ratio estimates. Robust standard errors, unstructured correlation structures and Wald *p*-values were used. Univariable analyses were performed using all cat and dog animal-months pooled, except when analyzing effects of anniversary timing (did or did not fall in the period when anniversary emails were not sent) and owner email status (email address listed as at 14 August 2017 or not). For these analyses, separate models were fitted for animals whose anniversary fell in the period when anniversary emails were not sent and animals whose anniversary did not fall in that period. Interactions between time period number from date of entry and owner email status were assessed within each subset of animals. Odds ratios were reported for cats and dogs pooled, other than where indicated.

## 3. Results

### 3.1. Animal Characteristics and Owner Update Patterns

In total, 394,747 cats and 904,909 dogs met the study inclusion criteria for describing animal characteristics (Table 1). Of the 1,299,656 animals implanted from 2008 to 2016 and registered on the CAR database which met the study inclusion criteria, more than twice as many dogs were registered than cats (70% of all study animals or 904,909 were dogs and 30% or 394,747 were cats) (Table 1). Sex was recorded for 99% of cats (391,642/394,747) and dogs (898,143/904,909). Of these, 50% (197,657/391,642) of cats and 49% (439,569/898,143) of dogs were female. There were minor differences in the proportions of sexes by state and territory (Appendix A). Date of birth was recorded for 93% (368,173/394,747) of cats and 94% (848,022/904,909) of dogs. Ages of these animals as at 17 August 2017 are shown in Figure 1 and Figure 2. Animals aged 0 to 8 years accounted for approximately 75% of registered animals, and 2.7% were recorded as >16 years.

#### 3.1.1. Age at Microchip Implantation

Distributions of ages at implantation are shown in Figure 3 and Figure 4. Cats were most frequently implanted at 3, 2, then 4 months of age, while dogs were most frequently implanted at 2, 3, then 4 months. Ninety-one percent (334,224) of cats were <5 years at microchip implantation, and 0.3% (1129) were >16 years, while 89% (755,952) of dogs were implanted at <5 years and 0.1% (1184) at >16 years.

#### 3.1.2. Breed

Of all animals implanted from 2008 to 2016 and registered on the CAR database, 81% of cats and 24% of dogs were mixed-bred. Purebred cats represented 19% of all cats registered and purebred dogs represented 76% of all dogs registered. For those registered in 2016, the five most numerous cat pure breeds were Ragdoll, Burmese, British, Bengal and Siamese. These breeds represented 66% of purebred cats and 12% of all cats, and Ragdolls were more than three times more numerous than the next most popular breed. The five most numerous dog pure breeds were Staffordshire Bull Terrier, Australian Kelpie, Labrador Retriever, Border Collie and Jack Russell Terrier. Together, these breeds represented 34% of purebred dogs and 26% of all dogs, with Staffordshires representing nearly 50% more than the next most popular breed (Table 3).

For animals implanted from 2008 to 2016 and registered on the CAR database, the Australian Capital Territory (ACT) had the highest proportions of both cats and dogs that were purebred (cats 34%; dogs 84%), SA had the lowest proportion of cats that were purebred (15%), and Western Australia (WA) had the lowest proportion of dogs that were purebred (70%) (detailed results not shown). The percentages of cats implanted by year from 2008 to 2016 and registered on the CAR database were purebred varied, with the highest in 2012 (21%) and lowest in 2015 (17%). Similarly for dogs, the highest percentage that was purebred was in 2008 (77%) and the lowest was in 2014 (75%).

#### 3.1.3. State/Territory

Victoria had the highest number of cats implanted per year from 2008 to 2016 and registered on the CAR database per 1000 residents, at an average of 5.34, while New South Wales (NSW) had the lowest at 0.12 (Appendix A). Tasmania had the highest number of dogs per 1000 residents per year at 12.0, and NSW had the lowest at 0.4, followed by Queensland for both cats and dogs (Appendix A). Of the total population of animals implanted from 2008 to 2016 and registered on the CAR database, Victoria had the greatest proportion of cats (70%) and dogs (61%), while the NT (0.8%) and ACT (0.9%) made up the smallest proportion of cats and dogs, respectively.

#### 3.1.4. Socioeconomic Index

Owners’ postcodes and postcode index of relative socioeconomic advantage and disadvantage were available for 99.8% (393,874/394,747) of cats and 99.7% (902,011/904,909) of dogs. Study animals’ postcodes reflected a wide range of socioeconomic statuses (high values represent areas of relative advantage). For cats and dogs combined, 29%, 23%, 23%, and 26% were from postcodes with index values <960, 960 to <1000, 1000 to <1040, and ≥1040, respectively. As at 2011, when index values were calculated, 27%, 21%, 20% and 33% of Australia’s population resided in postcodes with these index values.

### 3.2. Owner Details’ Updates

In total, 41,545 cats and 82,942 dogs were used in analyses of owner details updates; these were the cats and dogs newly registered from 1 March 2015 to 24 February 2016.

#### 3.2.1. Owner Updates in Relation to Registration and Anniversary Dates

Patterns of owner details updates are shown from registration dates (Figure 5 and Figure 6; Table 4, Table 5, Table 6 and Table 7). Updates of owner details were common on the day the animal was registered and the day after (days 0 and 1, respectively; Figure 5 and Figure 6). For the 41,545 cats and 82,942 dogs, 24% of cats (9784) and 20% of dogs (16,474) had updates of owner information on the day the animal was registered on the CAR system, and 43% of cats (17,689) and 37% of dogs (30,468) had update events on the following day. Owner updates were also relatively common from days 2 to 360 after date of registration (Figure 5 and Figure 6), with 35% (14,639/41,545) of cats and 43% of dogs (35,333/82,942) having at least one update of owner details in that period (Table 4). After the first three months from registration, daily numbers of animals with updates were highest in the days immediately following the sending of anniversary reminder emails (i.e., on days 367–371 after registration; Figure 5 and Figure 6). Similarly, numbers and proportions of animals with updates were highest during days 2–180 and 361–390 after registration date (Table 4). Odds of at least one update of owner details in the time period varied significantly by time from registration for cats and dogs combined (*p* < 0.001); relative to days 2–30, odds of updates were lower in every other time period; Table 4).

Of the 41,545 cats and 82,942 dogs used in analyses of owner details’ updates, no owner email address was listed in the CAR database as at 14 August 2017 for 30% (12,541) of cats and 31% (25,756) of dogs (Table 5). For 31% of cats (13,048) and 25% of dogs (20,767), the animal’s first anniversary fell in the period when anniversary email reminders were not sent. For animals whose first anniversary fell in the period when email reminders were sent, there was a significant interaction (*p* < 0.001) between owner email status and timing of updates (Table 5). This was partly due to a much larger increase in updates from days 361 to 390, compared to days 331 to 360, for owners with an email listed. For animals whose owners had an email address listed, the percentage of time periods where owner details were updated increased from 1.2% in days 331 to 360 to 2.3% in days 360 to 390 (odds ratio 2.04; 95% CI 1.87 to 2.23; *p* < 0.001). For animals whose owners did not have an email address listed in the CAR database as at 14 August 2017, the percentages of time periods where owner details were updated were 0.4% in days 331 to 360 and 0.6% in days 360 to 390, and the odds ratio for at least one update of owner details was much closer to 1 (odds ratio 1.25; 95% CI 0.98 to 1.58; *p* = 0.069). In contrast, for animals whose first anniversary fell in the period when email reminders were not sent, there were minimal or no increases in the percentage of time periods where at least one update of owner details occurred from days 331 to 360, compared to days 361 to 390. This applied to owners who had provided an email address but received no annual email reminder (1.4% to 1.3%), and to owners who had not provided an email address (0.5% to 0.7%). For cats and dogs combined, relative to days 331 to 360, the odds ratios for at least one update of owner details occurring from days 361 to 390 were 1.29 (95% CI 0.91 to 1.84; *p* = 0.152) where the owner had not provided an email address, and 0.91 (95% CI 0.77 to 1.06; *p* = 0.215) where the owner had provided an email address but received no email. For time periods of more than 90 days after registration, owner updates were approximately 2 to 3 times more likely for owners who provided an email address but did not receive an email reminder, compared to those who did not provide an email address or receive a reminder (Table 5).

Owners of older animals were less likely to have an email address listed than owners of younger animals. Within each of cats and dogs, percentages whose owner had an email address recorded declined progressively from 78% in those aged <1 year as at 17 August 2017 to 41% and 30%, respectively, in cats and dogs aged over 16 years (results not shown).

#### 3.2.2. Owner Updates by State/Territory

The probability of updates of owner information varied (*p* < 0.001) by state or territory (Table 6). Owners who resided in Victoria were most likely (3.4%) to update their details in any particular time period, and NT owners were least likely (1%). This was true for cats, dogs, and combined species.

#### 3.2.3. Owner Updates by Species and Age

Cats were less likely than dogs to have their owner’s details updated in any particular time period (OR 0.80, 95% CI 0.79 to 0.82; *p* < 0.001). In the 18 months from the registration date, for each animal (a total of 747,810 animal-months for cats and 1,492,956 animal-months for dogs), 2.5% of these months had at least one update event for cats and 3% for dogs.

Probabilities of at least one update of owner details in any particular time period also varied (*p* < 0.001) by the animal’s age at registration on the CAR database (Table 7). Increasing age was associated with decreasing probability of owner information updates for cats, dogs, and overall. Time periods for animals aged 10 years or more at registration had the least probability of an update (0.8% to 1%) and time periods for animals aged 1 year at registration had the highest probability of an update (3.2%).

#### 3.2.4. Owner Updates by Socioeconomic Index

Probabilities of at least one update of owner information in any particular time period varied by the socio-economic status of the owner’s postcode (*p* < 0.001; Table 8). Owners from postcodes with the highest socioeconomic indices were most likely to update their details and owners from the lowest indices were least likely for cats, dogs and overall. For example, for animals with owners from postcodes with the lowest socioeconomic index value, owner information was updated in 2.4% of the time periods, compared to animals with owners from postcodes with the highest index values, who updated in 3.2% of time periods.

## 4. Discussion

In this study, we analyzed the data for 394,747 cats and 904,909 dogs registered with Central Animal Records, Australia’s largest national companion animal microchip database, to describe the characteristics of these cats and dogs and to determine whether sending email reminders increased the frequency that owners updated their details on the database. Key findings from this study were that sending annual email reminders increased the frequency that owners updated their information, and the frequency of update events varied by state or territory, animal species, pet age and socioeconomic status of the owner’s locality.

### 4.1. Characteristics of Registered Animals

#### 4.1.1. Species

Of the cats and dogs implanted from 2008 and 2016 and registered with CAR, more were dogs (70%) than cats (30%). This is not surprising, considering that there are more pet dogs (4.2 million) than cats (3.3 million) in Australian households and more of these dogs (76%) are microchipped than cats (64%) [1]. In contrast, only 9% of cats entering RSPCA Queensland shelters were microchipped, while 28% of dogs had a microchip [27]. This highlights the issue that fewer cats than dogs are microchipped, and cats are at a higher risk of euthanasia than dogs when they cannot be reunited with owners after straying [2,4,10,27,28]. Owners with indoor-only cats might not perceive identification with a microchip as important, but 28% to 40% of lost cats were described by their owners as being confined indoors [38]. Semi-owned cats probably also contribute substantially to the low proportion of stray cats admitted to shelters that are microchipped. For example, in a study of admissions to RSPCA shelters, for 33% of stray cat admissions, the person surrendering had provided care for more than one month but did not perceive themselves to be the owner [39]. These semi-owned cats are thought to constitute a large proportion of shelter admissions [40,41,42].

#### 4.1.2. Sex Differences

Although the differences were minimal, male dogs (51%) implanted from 2008 to 2016 and registered with CAR outnumbered females (49%), but there were equal proportions of male and female cats. Studies show that males are more likely to be microchipped than females [27,28] and that male animals (51–54%) are marginally overrepresented in Australian shelters [6,27].

#### 4.1.3. Age at Implantation

Most animals (91% of cats, 89% of dogs) were <5 years of age at the time of microchip implantation. Of these, cats were most frequently implanted at 3, 2 and 4 months of age, while dogs were most frequently implanted at 2, 3 and 4 months, respectively. This corresponds with state and territory microchipping legislation, where cats and dogs are required to be implanted and recorded on a database by 12 weeks of age for the ACT, NSW, Queensland and Victoria, and six months for Tasmania and WA (Appendix A) [11,13,14,15,16,17,18,19,20]. The remainder of the study population was implanted older than five years, which could coincide with microchipping prior to transfer of ownership. For states and territories where compulsory microchipping has only been recently introduced, animals born prior to the introduction date are required to be microchipped prior to transfer of ownership [11,13,14,15,16,19,20]. These results suggest that many owners are complying with compulsory microchipping laws.

#### 4.1.4. Breed Differences

More of the cats implanted with a microchip between 2008 and 2016 and registered on the CAR database were mixed-bred (81%) than purebred, while a much lower proportion of the dogs registered were mixed-bred (24%). Purebred dogs comprised 52% of all Australian dogs in 2016 [43], and purebred animals are more likely than mixed-breds to be microchipped [28]. Thus, mixed-bred dogs were underrepresented on the CAR database but are overrepresented in Australian and US shelters [6,36]. An explanation for the disparity between proportions of total Australian mixed-bred dogs (48%), CAR mixed-breds (24%), and overrepresentation in shelters (72–83%) may be that dogs from unwanted litters are more likely to be mixed-bred and not microchipped when given away. The proportions of cats and dogs that were purebred decreased slightly from 2008 to 2016, which may reflect a greater public awareness of the need to adopt animals to reduce shelter euthanasia percentages [44,45,46].

The five most numerous pure breeds of cats newly registered with CAR were Ragdoll, Burmese, British, Bengal and Siamese. The five most commonly registered pure dog breeds were Staffordshire Bull Terrier, Australian Kelpie, Labrador Retriever, Border Collie and Jack Russell Terrier. This was mirrored in the most common pure dog breeds processed through RSPCA Queensland shelters, where Staffordshire Bull Terriers were the most prevalent, Australian Kelpies were equal ninth, Labrador Retrievers were second, Border Collies were sixth, and Jack Russell Terriers were equal ninth [6], and in Australian National Kennel Council figures where Staffordshire Bull Terriers were the second most popular dog breed, Labrador Retrievers were most prevalent and Border Collies were fifth (Australian Kelpies and Jack Russell Terriers were not within the top 10 breeds) [47]. Given that the most common dog breeds admitted to RSPCA shelters closely matched the most common breeds registered on the CAR database, this suggests that these breeds are overrepresented in shelters because there are proportionately more of them in the Australian dog population, as opposed to breed characteristics (for example behavioral, health) increasing their risk of admission to shelters.

#### 4.1.5. State/Territory

The number of new animals implanted with a microchip and registered with CAR per 1000 residents per year generally corresponded with state and territory legislation. Although the largest, CAR is one of six microchip registries in Australia [21,22,23,24,25,26] and the number of study animals per 1000 residents in each state or territory is not a true reflection of actual animal populations. States with some of the oldest microchipping legislation [17,18,48] had the highest numbers of cats and dogs registered per 1000 residents per year (Victoria, 5.34 cats; Tasmania, 12 dogs). These states are also known to have the highest percentages of dogs and cats microchipped [10] and highest frequencies of pet ownership in Australia [43]. NSW had the lowest numbers of new registrations in the CAR database per 1000 residents per year (0.12 cats, 0.4 dogs). All cats and dogs residing in NSW are legally required to have their microchip registered with the New South Wales Pet Registry, the state’s government database [14], and owners are not required to register their pets secondarily with CAR. Queensland and ACT had the second and third lowest numbers of cat and dog registrations per 1000 residents (Queensland, 1.09 cats, 2.73 dogs; ACT 1.1 cats, 2.43 dogs). The ACT has previously been reported to have low percentages of cats registered with municipal council and microchipped [49]. Queensland is likely to have low numbers of registrations with CAR because of the competing database, Home Safe ID, which services Queensland branches of the RSPCA and Animal Welfare League. In addition, Queensland introduced compulsory microchipping later, in 2008, compared to 2005 in Victoria [15,16].

### 4.2. Email Reminders and Updates of Owner Information

Email reminders increased the frequency that owners updated their pets’ microchip details. Nearly twice as many animals had owner information updated immediately after emails were sent than in the corresponding period when emails were not sent (2.3% versus 1.3%). Owners who received an email also updated their details more frequently (2.3%) than owners with no email address listed (0.6%). While our results provide strong evidence that email reminders result in an immediate increase in frequency of owner details updates, this increase appears to be modest relative to the frequency of updates occurring in months unrelated to email reminders; there was also continual updating of owner details throughout the study period, not associated with email reminders. This continual low level of update events is likely to be due to individual owner diligence, and due to councils, veterinarians and other animal workers promoting updating of microchips [31,32,33,35].

It is possible that some of the immediate increase in frequency of owner details updates due to email reminders is because they bring forward some updates that would otherwise have occurred in subsequent months. It is difficult to ascertain the extent of this from our data. If this is common, we would expect the frequency of updates to be less in the months after the email reminder month compared to immediately before the email reminder. For animals whose first registration anniversary did not fall on or between 31 March 2016 and 4 July 2016 and whose owner had recorded an email address, there was little evidence of this pattern. In the five months after the email reminder month (361–390 days), percentages of animals with updates were 1.6%, 1.4%, 1.1%, 1.1% and 1.2%; these were similar to or greater than the percentages in the five months preceding the email reminder month of 1.2% to 1.4% (Table 5). Bringing forward updates of owner details is beneficial, as the associated animal could stray in the interim. However, bringing forward updates is of less benefit than stimulating owner details updates that would otherwise not have occurred at all. However, our results show that email reminders are useful. Assuming they are cheap and easy to send, we recommend them as part of a set of strategies to increase the ongoing accuracy and completeness of owner details on pet microchip databases.

In total, 31% of cats and 27% of dogs had an update of owner information between 1 March 2015 and 17 August 2017. Despite this, a large proportion (37%) of microchipped animals presented to RSPCA Queensland shelters had incorrect owner details [27]. By using these analyses, microchip databases and municipalities worldwide could implement similar strategies to increase update compliance and investigate other reminder systems for owners who do not have an email address or access to the internet (for example, Short Message Service (SMS) reminders). An additional method to increase correct owner information could be for microchip databases to collaborate with energy providers, whereby owners could give permission for their new address details to be shared at the time energy is connected.

### 4.3. Updates of Owner Information According to Animal Characteristics

#### 4.3.1. Owner Updates by State/Territory

Updates of owner details were more common for pets whose owners resided in Victoria (3.4% of animal-months), and least common for pets with NT owners (1% of animal-months). Frequency of owner details updates may be influenced in some way by compulsory microchipping, but further research is required to identify reasons for differences by state/territory. At the time of writing, SA and NT were the only state and territory that had no legal requirement for compulsory microchipping, while all remaining states and territories mandated microchipping and registration on a database by a specified age (12 weeks or six months) or before change of ownership (Appendix A) [11,12,13,15,16,17]. Microchipping was not mandated at a state or territory level in NT or SA, but was required in one NT local government area (i.e., one municipal council [50]) and several SA municipal councils [51]. This could explain why NT owners updated their details less frequently than SA (2.4%) and other states and territories.

#### 4.3.2. Owner Updates by Species

Cats (2.5%) had less frequent updates of owner details than dogs (3%; *p* < 0.001). A study of stray animals entering RSPCA Queensland found no difference in microchip data problems between the two species at intake; for both species, 63% had correct details in 2012–2013) [27]. However, in this same financial year, 35% of Queensland dogs (37% nationally) but only 4.3% (4.6% nationally) of cats entering RSPCA shelters were reclaimed [52]. This indicates that factors other than ability to contact owners are associated with reclaiming of pets, and that increasing the accuracy of owner details in microchip databases may have little influence on percentages of stray cats that are reclaimed. However, in Victoria, 32/79 councils have achieved reclaim rates of 10% or higher, with some over 30% for cats, based on website data on their individual Domestic Animal Management Plans [53]. These results also highlight the potential for further studies into why cats have lower rates of microchipping and updates of owner information. For example, to investigate if provision of secure facilities for cats or assisting owners with transport of their cats when community microchip events are held increases the frequency of microchipping and updating of owner contact details.

#### 4.3.3. Owner Updates by Animal Age

Younger animals had their owner’s details updated more frequently than older animals (e.g., 3.2% of animal-months for animals in their first year of life; 1.1% for animals aged 10 years or more). However, a substantial proportion of dogs in shelters are older animals [6]. Possible reasons for the decline in update frequency as pets age are that owners may forget to update their details because of the greater time elapsed since implantation, the animal is deceased, and details of owners of older animals may change less frequently than for owners of younger animals. In addition, owners of older animals were less likely to have an email address listed. This may explain why the update frequency for owners who provided an email address but received no annual email reminder was approximately double that of owners who had not provided an email address (1.4% versus 0.5% for days 331–360; and 1.3% versus 0.7% for days 361–390). By highlighting the lower frequency of updates as animals age, this finding could provide the foundation for targeted campaigns and advertising aimed at encouraging updates of microchip details in aged cats and dogs.

#### 4.3.4. Owner Updates by Socioeconomic Area

Owners from postcodes with the highest socioeconomic indices updated their pets’ microchip details more frequently (3.2% of animal-months) than owners from the lowest indices (2.4% of animal-months). It is known that lower socioeconomic areas contribute more to shelter intake numbers [6,48], and socioeconomic status is positively correlated to reclaim percentages in USA [10]. Potential explanations for poor compliance with updating details in low socioeconomic populations are decreased access to phone and internet (as considered when assigning an index of relative socioeconomic advantage and disadvantage [37,54]), and costs associated with changing some microchip details [21]. A UK survey revealed that owners’ reasons for not updating their pets microchip details included fees incurred for updating details and a complicated process to change their details [30]. By demonstrating that owners in lower socioeconomic postcodes update their details less frequently but contribute more animals to shelters, this can provide a basis for better allocation of resources in these municipalities to promote increased owner awareness and opportunities to update, including free and low-cost microchipping events.

### 4.4. Limitations

We probably underestimated the strength of the relationship between owners receiving an email reminder and updating their details on the CAR database. Firstly, when owners with >1 to ≤5 pets with different registration dates updated their information upon receiving an annual email reminder for one of their pets, these updates would be applied to all pets, yet the update may not have occurred soon after those other animals’ anniversary dates. Secondly, we assumed owner email statuses at the time of their pets’ anniversaries were the same as at 17 August 2017, whereas it is likely that some owners with email access at that date did not, in fact, have email access earlier, at the time of their pets’ anniversaries. Furthermore, we classified text strings containing the ‘@’ symbol as email addresses but were not able to assess the validity of those email addresses. Thirdly, it is likely that some animal-months included in the study were for animals after they had died. Annual email reminders ceased once the animal was recorded on the database as deceased. Therefore, our analyses would show that these owners received an email but did not update their details in the following time period, when in fact they did not receive an email. More commonly, owners did not record on the database that their pet had died, and owners of deceased animals would be unlikely to respond to an email reminder. However, the impacts of deceased animals on our estimated effects of email reminders by time from registration were probably small, as relatively few animals would be expected to have died within 540 days (i.e., 18 months) of their registration date.

## 5. Conclusions

Email reminders increase the frequency that owners update their details on their pets’ microchip database. We recommend them as part of a set of strategies to increase the ongoing accuracy and completeness of owner details. The frequency of updates of owner details also varies by state or territory, pet species and age, and socioeconomic index of the owner’s postcode. This study highlights the need for targeted campaigns aimed at encouraging updating contact details for owners who were shown to update least frequently: cat owners, owners of older animals, and owners in low socioeconomic municipalities. By increasing pet owner compliance in updating their details on microchip databases, this is likely to improve reclaim percentages of stray animals and, subsequently, reduce the number of pets euthanized in shelters and pounds each year.

## Figures and Tables

**Figure 1 animals-08-00020-f001:**
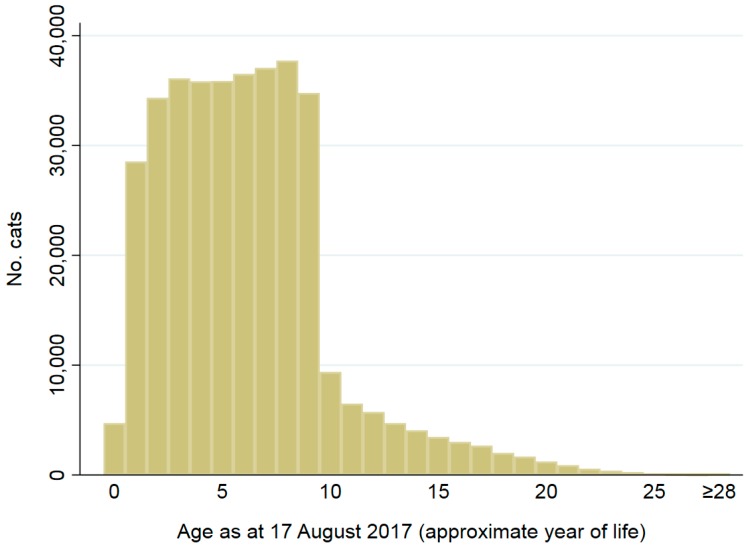
Ages (approximate year of life) as at 17 August 2017 for 368,173 cats implanted from 2008 to 2016 and registered on the CAR database. Year of life was approximated as 0 if the cat was aged ≤365 days, 1 if the cat was aged ≤730 days (i.e., 365 + 365 days), 2 if the cat was aged ≤1095 days (i.e., 365 + 365 + 365 days, 3 if the cat was aged ≤1461 days (i.e., 365 + 365 + 365 + 366 days), 4 if the cat was aged ≤1826 days (i.e., 365 + 365 + 365 + 366 + 365 days), and so on.

**Figure 2 animals-08-00020-f002:**
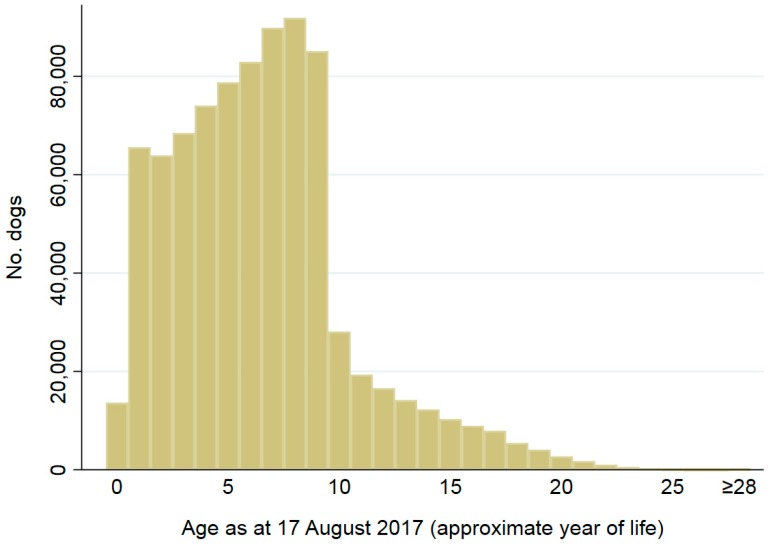
Ages (approximate year of life) as at 17 August 2017 for 848,022 dogs implanted from 2008 to 2016 and registered on the CAR database. Year of life was approximated as 0 if the dog was aged ≤365 days, 1 if the dog was aged ≤730 days (i.e., 365 + 365 days), 2 if the dog was aged ≤1095 days (i.e., 365 + 365 + 365 days, 3 if the dog was aged ≤1461 days (i.e., 365 + 365 + 365 + 366 days), 4 if the dog was aged ≤1826 days (i.e., 365 + 365 + 365 + 366 + 365 days), and so on.

**Figure 3 animals-08-00020-f003:**
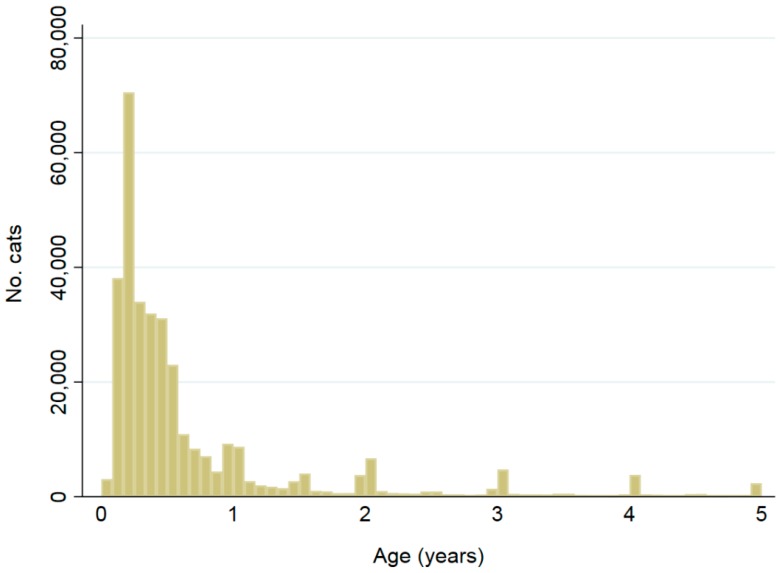
Ages at implantation date for 334,224 cats ≤5 years of age; a further 33,954 cats had recorded ages >5 years including 1129 (0.3%) recorded as having been implanted when aged ≥16 years. Bar widths are 1/12 of a year.

**Figure 4 animals-08-00020-f004:**
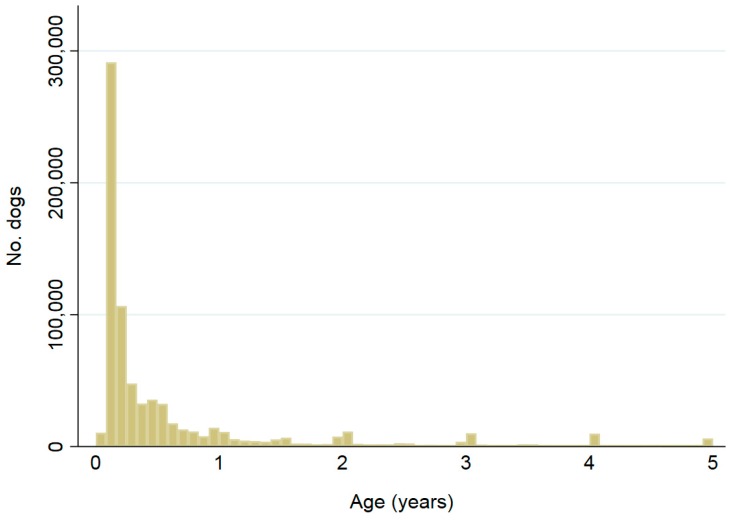
Ages at implantation date for 755,952 dogs ≤5 years of age; a further 92,078 dogs had recorded ages >5 years including 1184 (0.1%) recorded as having been implanted when aged ≥16 years. Bar widths are 1/12 of a year.

**Figure 5 animals-08-00020-f005:**
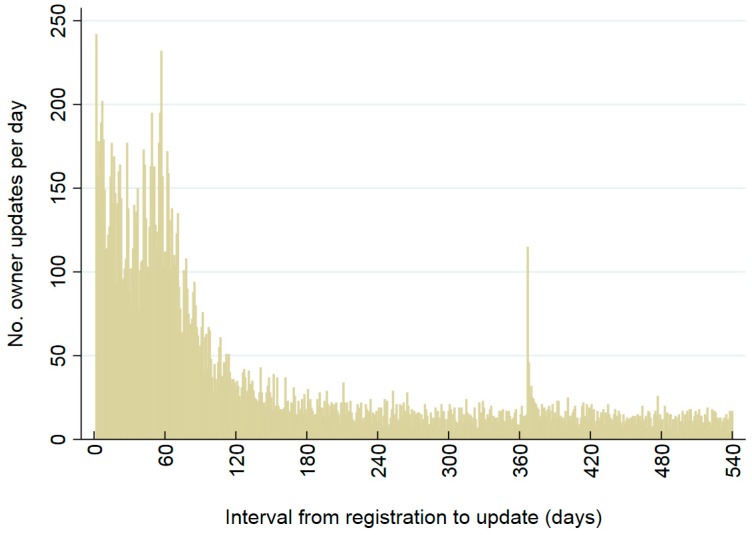
Numbers of 41,545 cats whose owner’s details were updated, by day since registration on the CAR system. Updates on day of registration (day 0) and day 1 were excluded. Owners who were and were not sent email reminders are included.

**Figure 6 animals-08-00020-f006:**
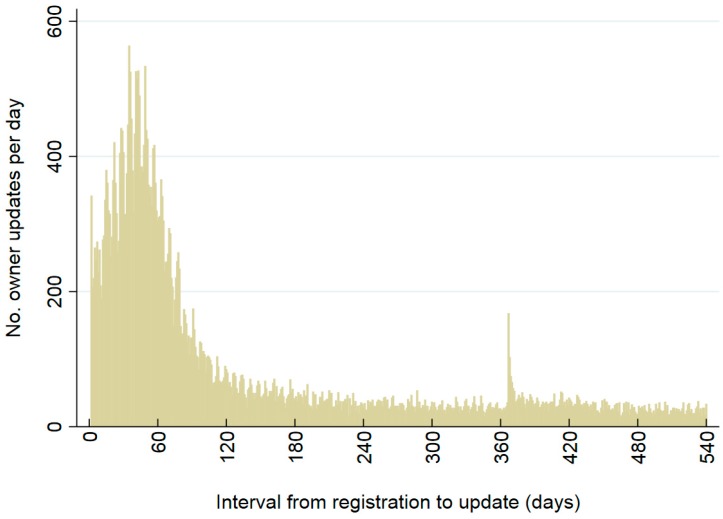
Numbers of 82,942 dogs whose owner’s details were updated, by day since registration on the CAR system. Updates on day of registration (day 0) and day 1 were excluded. Owners who were and were not sent email reminders are included.

**Table 1 animals-08-00020-t001:** Numbers of cats and dogs registered on the Central Animal Records microchip database and included in the study, by year of microchip implantation.

Year of Implantation	Cats	Dogs	Total
2008	46,135	119,075	165,210
2009	48,846	121,716	170,562
2010	43,757	116,732	160,489
2011	40,647	109,018	149,665
2012	41,442	97,443	138,885
2013	45,476	88,456	133,932
2014	42,388	82,060	124,448
2015	42,489	84,168	126,657
2016	43,567	86,241	129,808
Total	394,747	904,909	1,299,656

**Table 2 animals-08-00020-t002:** Explanations and sizes of study populations, according to type of analysis.

Analysis	Study Population	Numbers of Animals Included	Other
Animal characteristics	Cats and dogs with microchip implanted from 1 January 2008 to 31 December 2016	394,747 cats and 904,909 dogs	
Updates of owner details	Cats and dogs first registered with CAR from 1 March 2015 to 24 February 2016	41,545 cats and 82,942 dogs	Updates from 1 March 2015 to 17 August 2017

**Table 3 animals-08-00020-t003:** Top 20 cat and dog pure breeds registered on the Central Animal Records microchip database in 2016. The numbers of animals belonging to each breed, and percentages of newly registered animals each breed comprised are presented.

Rank	Cats	No.	% of All Cats Registered in 2016 (*n* = 43,567)	Dogs	No.	% of All Dogs Registered in 2016 (*n* = 86,241)
1	Ragdoll	2702	6.20%	Staffordshire Bull Terrier	6138	7.12%
2	Burmese	833	1.91%	Australian Kelpie	4306	4.99%
3	British	674	1.55%	Labrador Retriever	3979	4.61%
4	Bengal	425	0.98%	Border Collie	3936	4.56%
5	Siamese	412	0.95%	Jack Russell Terrier	3902	4.52%
6	Russian	358	0.82%	German Shepherd	2726	3.16%
7	Tonkinese	279	0.64%	Poodle	2095	2.43%
8	Birman	256	0.59%	Chihuahua	2016	2.34%
9	Devon Rex	255	0.59%	Australian Cattle Dog	1969	2.28%
10	Persian	224	0.51%	American Staffordshire Terrier	1832	2.12%
11	Maine Coon	190	0.44%	Maltese	1804	2.09%
12	Scottish Fold	110	0.25%	Golden Retriever	1747	2.03%
13	Exotic	86	0.20%	Cavalier King Charles Spaniel	1526	1.77%
14	Australian Mist	77	0.18%	Rottweiler	1309	1.52%
15	Siberian	77	0.18%	Dachshund	1246	1.44%
16	Sphynx	77	0.18%	Pug	1038	1.20%
17	Manx	74	0.17%	French Bulldog	999	1.16%
18	Abyssinian	71	0.16%	Fox Terrier	955	1.11%
19	Oriental	67	0.15%	Bulldog	894	1.04%
20	Chinchilla	57	0.13%	Siberian Husky	854	0.99%

**Table 4 animals-08-00020-t004:** Percentages of animals that had at least one update of owner details in each time period, by time from the animal's date of registration on the CAR database. *n* refers to the numbers of animals analyzed.

Time Period (Days from Date of Registration)	% with at Least One Owner Details Update in Time Period			Odds Ratio	95% Confidence Interval (CI)	*p*
	Cats (*n* = 41,545)	Dogs (*n* = 82,942)	Combined (*n* = 124,487)			
2 to 30	9.7%	10.4%	10.2%	Reference group		
31 to 60	9.5%	14.5%	12.8%	1.29	1.26 to 1.33	<0.001
61 to 90	6.6%	7.6%	7.3%	0.69	0.67 to 0.71	<0.001
91 to 120	3.2%	3.3%	3.3%	0.30	0.29 to 0.31	<0.001
121 to 150	2.1%	2.1%	2.1%	0.19	0.18 to 0.20	<0.001
151 to 180	1.5%	1.7%	1.6%	0.15	0.14 to 0.15	<0.001
181 to 210	1.3%	1.4%	1.4%	0.13	0.12 to 0.13	<0.001
211 to 240	1.1%	1.2%	1.2%	0.10	0.10 to 0.11	<0.001
241 to 270	1.1%	1.2%	1.2%	0.10	0.10 to 0.11	<0.001
271 to 300	1%	1.1%	1%	0.09	0.09 to 0.10	<0.001
301 to 330	1%	1%	1%	0.09	0.08 to 0.09	<0.001
331 to 360	0.9%	1%	1%	0.09	0.08 to 0.09	<0.001
361 to 390	1.6%	1.6%	1.6%	0.14	0.14 to 0.15	<0.001
391 to 420	1%	1.2%	1.2%	0.10	0.10 to 0.11	<0.001
421 to 450	0.9%	1.1%	1.1%	0.09	0.09 to 0.10	<0.001
451 to 480	0.9%	0.9%	0.9%	0.08	0.08 to 0.09	<0.001
481 to 510	0.9%	0.9%	0.9%	0.08	0.07 to 0.08	<0.001
511 to 540	0.9%	0.9%	0.9%	0.08	0.07 to 0.08	<0.001
2 to 360	35.2%	42.6%	40.1%			
2 to 540	38.9%	46.5%	43.9%			

**Table 5 animals-08-00020-t005:** Percentages of animals with at least one update of owner details, by time period from the animal’s date of registration on the CAR database, according to first anniversary date and owner email address status ^1^. *n* refers to the numbers of animals analyzed.

Time Period (Days from Date of Registration)	Cats		Dogs		Combined	
	Email Address Not Provided	Email Address Provided	Email Address Not Provided	Email Address Provided	Email Address Not Provided	Email Address Provided
First anniversary of registration did not fall on or between 31 March 2016 and 4 July 2016; anniversary emails sent to owners with email addresses recorded						
	(*n* = 8481)	(*n* = 20,016)	(*n* = 19,155)	(*n* = 43,020)	(*n* = 27,636)	(*n* = 63,036)
2 to 30	10.6%	12.5%	8%	12.6%	8.8%	12.6%
31 to 60	7.3%	10.1%	9.7%	17%	9%	14.8%
61 to 90	4%	6.1%	3.9%	7.7%	4%	7.2%
91 to 120	2.3%	3.1%	1.8%	3.4%	1.9%	3.3%
121 to 150	1.5%	2.3%	1.2%	2.3%	1.3%	2.3%
151 to 180	0.9%	1.7%	0.9%	2%	0.9%	1.9%
181 to 210	0.9%	1.7%	0.9%	1.7%	0.9%	1.7%
211 to 240	0.6%	1.3%	0.8%	1.4%	0.7%	1.4%
241 to 270	0.8%	1.3%	0.7%	1.4%	0.7%	1.3%
271 to 300	0.4%	1.1%	0.6%	1.3%	0.5%	1.2%
301 to 330	0.6%	1.1%	0.6%	1.2%	0.6%	1.1%
331 to 360	0.4%	1.1%	0.4%	1.2%	0.4%	1.2%
361 to 390	0.6%	2.3%	0.5%	2.4%	0.6%	2.3%
391 to 420	0.6%	1.3%	0.5%	1.7%	0.5%	1.6%
421 to 450	0.4%	1.2%	0.4%	1.5%	0.4%	1.4%
451 to 480	0.4%	1.2%	0.4%	1.1%	0.4%	1.1%
481 to 510	0.4%	1.2%	0.4%	1.1%	0.4%	1.1%
511 to 540	0.5%	1.2%	0.4%	1.2%	0.4%	1.2%
First anniversary of registration fell on or between 31 March 2016 and 4 July 2016; anniversary emails not sent						
	(*n* = 4060)	(*n* = 8988)	(*n* = 6601)	(*n* = 14,166)	(*n* = 10,661)	(*n* = 23,154)
2 to 30	4.4%	5.1%	6.5%	9%	5.7%	7.5%
31 to 60	7.5%	11%	10.1%	15.1%	9.1%	13.5%
61 to 90	6.2%	10.5%	7.1%	12.1%	6.8%	11.4%
91 to 120	2.7%	4.4%	2.9%	5.3%	2.8%	5%
121 to 150	1.1%	2.5%	1.7%	2.8%	1.5%	2.7%
151 to 180	0.8%	1.9%	1.2%	2.1%	1.1%	2.1%
181 to 210	0.7%	1.2%	0.6%	1.9%	0.7%	1.6%
211 to 240	0.7%	1.4%	0.7%	1.5%	0.7%	1.4%
241 to 270	0.6%	1.2%	0.9%	1.4%	0.8%	1.3%
271 to 300	0.6%	1.3%	0.5%	1.4%	0.5%	1.3%
301 to 330	0.8%	1.2%	0.6%	1.4%	0.7%	1.3%
331 to 360	0.6%	1.2%	0.5%	1.5%	0.5%	1.4%
361 to 390	0.6%	1.1%	0.7%	1.3%	0.7%	1.3%
391 to 420	0.7%	0.9%	0.6%	1.2%	0.6%	1.1%
421 to 450	0.5%	0.8%	0.6%	1.2%	0.6%	1%
451 to 480	0.3%	1.0%	0.6%	1.3%	0.5%	1.2%
481 to 510	0.3%	0.9%	0.5%	1.1%	0.4%	1%
511 to 540	0.3%	0.8%	0.4%	0.9%	0.4%	0.9%

^1^ Owner email address status according to whether or not an email address was listed in the CAR database as at 14 August 2017.

**Table 6 animals-08-00020-t006:** Percentages of animal-months where the animal had at least one update of owner details, by owner state or territory. *n* refers to the number of animal-months in the 18 months from each animal’s date of registration on the CAR database.

Owner’s State or Territory ^1^	Cats		Dogs		Combined		Odds Ratio (OR)	95% CI	*p*
	*n*	%	*n*	%	*n*	%			
ACT	9306	1.2%	15,102	1.5%	24,408	1.4%	Reference group		
NSW	13,392	2.1%	43,902	3.3%	57,294	3.1%	2.51	2.21 to 2.85	<0.001
NT	10,116	0.9%	28,512	1.1%	38,628	1%	0.60	0.51 to 0.71	<0.001
QLD	65,520	1.5%	162,288	2.3%	227,808	2%	1.44	1.27 to 1.62	<0.001
SA	23,832	1.5%	80,658	2.6%	104,490	2.4%	1.96	1.73 to 2.23	<0.001
TAS	12,222	1.5%	61,956	1.4%	74,178	1.4%	0.93	0.81 to 1.07	0.338
VIC	567,162	2.8%	882,360	3.7%	1,449,522	3.4%	2.81	2.50 to 3.17	<0.001
WA	46,242	1.5%	218,178	1.3%	264,420	1.4%	1.01	0.89 to 1.14	0.910

^1^ ACT: Australian Capital Territory; NSW: New South Wales; NT: Northern Territory; QLD: Queensland; SA: South Australia; TAS: Tasmania; VIC: Victoria; WA: Western Australia.

**Table 7 animals-08-00020-t007:** Owner information updates by animal’s age at registration on the CAR database. *n* refers to the number of animal-months in the 18 months from each animal's date of registration on the CAR database; % refers to the percentage of animal-months that had at least one owner information update. Animals older than 20 years were excluded.

Animal’s Age at Registration (Approximate Year of Life and Days in Brackets)	Cats		Dogs		Combined		Odds Ratio	95% CI	*p*
	*n*	%	*n*	%	*n*	%			
1 (0 to 365)	526,428	2.6%	1,077,840	3.5%	1,604,268	3.2%	Reference group		
2 (366 to 730)	68,256	2.3%	75,438	1.6%	143,694	1.9%	0.50	0.48 to 0.52	<0.001
3 (731 to 1095)	30,924	2.8%	44,622	1.7%	75,546	2.1%	0.57	0.54 to 0.60	<0.001
4 (1096 to 1461)	17,496	2.5%	36,396	1.5%	53,892	1.8%	0.47	0.44 to 0.51	<0.001
5 (1462 to 1826)	11,232	2.1%	33,390	1.2%	44,622	1.5%	0.36	0.33 to 0.40	<0.001
6 (1827 to 2191)	9540	2.3%	27,180	1.2%	36,720	1.5%	0.39	0.36 to 0.43	<0.001
7 (2192 to 2556)	5508	1.9%	20,484	1.2%	25,992	1.3%	0.34	0.30 to 0.38	<0.001
8 (2557 to 2922)	5238	1.8%	18,900	1.1%	24,138	1.3%	0.32	0.28 to 0.36	<0.001
9 (2923 to 3287)	5148	1.7%	16,398	0.9%	21,546	1.1%	0.28	0.24 to 0.32	<0.001
10 (3288 to 3652)	3348	1.9%	14,832	0.8%	18,180	1%	0.24	0.19 to 0.29	<0.001
11 (3653 to 4017)	3870	1.7%	17,316	0.9%	21,186	1%	0.27	0.23 to 0.32	<0.001
12 (4018 to 4383)	2736	1.2%	10,080	0.7%	12,816	0.8%	0.18	0.14 to 0.23	<0.001
13 to 20 (4384 to 7305)	8874	1.1%	26,730	0.8%	35,604	0.8%	0.21	0.18 to 0.24	<0.001

**Table 8 animals-08-00020-t008:** Percentages of animal-months with at least one update of owner information, by socioeconomic status of owner postcode as at 2011 [37]. *n* refers to the number of animal-months in the 18 months from each animal's date of registration on the CAR database.

Index of Relative Socio-Economic Advantage and Disadvantage of Owner’s Postcode	Cats		Dogs		Combined		Odds Ratio	95% CI	*p*
	*n*	%	*n*	%	*n*	%			
<960	186,120	2.1%	400,788	2.6%	586,908	2.4%	Reference group		
960 to <1000	183,294	2.5%	380,952	2.7%	564,246	2.6%	1.14	1.11 to 1.16	<0.001
1000 to <1040	166,914	2.7%	325,728	3.2%	492,642	3%	1.32	1.29 to 1.36	<0.001
≥1040	208,710	2.7%	377,244	3.4%	585,954	3.2%	1.44	1.41 to 1.48	<0.001

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
