# Peer review of "Email Reminders Increase the Frequency That Pet Owners Update Their Microchip Information"

_animals, 2018, doi:10.3390/ani8020020_

Round 1

Reviewer 1 Report

Specific comments:

Line 67 – remove ‘some’ in “in a 2009 study of some USA shelters”

Line 120-121 – add words to increase clarity – “especially which breeds and age groups are overrepresented in shelters”

Line 154-155 – do you know how many owners were excluded because there was no email associated?

Line 158 – remove unnecessary ‘;’

Line 157-158 – it would be ideal to clarify the time periods in this sentence; when email reminders were and were not sent.

Section 2.5 is quite hard to follow – perhaps consider a table to show the different groups of owners and date ranges? Perhaps giving each group a defined consistent term that makes the subsequent data easier to follow – e.g. ‘reminded owners’, ‘non-email owners’, etc

Consider presenting table 2 as age histograms; it’s a large table of data to view only as numbers.

Figure 3 and 4 are excellent – display really clear peaks in registration coinciding with reminder emails. But are these both owners that were sent reminders and were not? If this includes both types of owners, perhaps their data could be colour coded to show both populations on the one graph? This may become too difficult to read, so is only a suggestion. Would be good to state what population of owners is included in the figure titles (i.e. received a reminder or did not).

Line 339 – space missing ahead of “In contrast, for animals...”

Consider a restructure of the results – pairing up the descriptive stats with the odds ratios – e.g. 3.2.4 immediately following 3.1.4, so that the whole question of socio-economics is dealt with in one go. This is only a suggestion, the authors may have good reason to stick with the current structure.

For consideration for section 4.2: Owners without emails – is updating difficult for them? Perhaps updating online is more straightforward when you have an email address logged with the system. i.e. getting reminders is not the only issue with non-email owners, perhaps these are also non-internet users for which the updating process is more cumbersome? The fact that owner updates were approximately 2 to 3 times more likely for owners who provided an email address but did not receive an email reminder, compared to those who did not provide an email address or receive a reminder suggests this might be the case? Or people who don’t supply an email address move less frequently?

For consideration for section 4.3.1: Difference in frequency of updates between locations could be because people move more frequently in one place (e.g. Victoria) as compared to another (e.g. NT)?

Line 523 – is this difference between cats and dogs significant? 

Author Response

The authors would like to thank you for taking the time to read our manuscript and provide their suggestions and recommendations. Below are our responses to your comments:

Reviewer 1

Line 67 – remove ‘some’ in “in a 2009 study of some USA shelters”

Amended

Line 120-121 – add words to increase clarity – “especially which breeds and age groups are overrepresented in shelters”

Added specifics in line 127-129:

·         For example, studies from several first world nations show that adult pets, mixed-bred animals, and purebred Staffordshire bull terriers are the most numerous categories in shelters[6, 9, 28, 36].

Line 154-155 – do you know how many owners were excluded because there was no email associated?

This section describes CAR's process, rather than the study process, i.e. rather than numbers of study animals classified as having owners with no email address. However, we have included study population sizes in the newly-added Table 2. We have also added numbers of cats and dogs in Results.

Line 158 – remove unnecessary ‘;’

Amended

Line 157-158 – it would be ideal to clarify the time periods in this sentence; when email reminders were and were not sent.

Sentence changed to read (line 169-170):

·         The updating of owner details when emails were sent was compared to the period when emails were not sent (emails were not sent from 31 March 2016 to 4 July 2016).

Section 2.5 is quite hard to follow – perhaps consider a table to show the different groups of owners and date ranges? Perhaps giving each group a defined consistent term that makes the subsequent data easier to follow – e.g. ‘reminded owners’, ‘non-email owners’, etc

Section about population subset 2 (with estimated anniversary dates) has been removed as this data was not used in owner update patterns analysis in the current version of the paper.

Table 2 (line 161) has been added to clarify time frames:

Table 2. Explanations and sizes of study populations, according to type of analysis.

Analysis

Study population

Numbers of animals included

Other

Animal characteristics

Cats and dogs with microchip implanted   from 1 January 2008 to 31 December 2016

394,747 cats and 904,909 dogs

Updates of owner details

Cats and dogs first registered with CAR   from 1 March 2015 to 24 February 2016

41,545 cats and 82,942 dogs

Updates from 1 March 2015 to 17 August   2017

Sections 2.1-2.6 have also been reworded to clarify the study design.

Consider presenting table 2 as age histograms; it’s a large table of data to view only as numbers.

Figures 1 and 2 have been added to replace the original Table 2:

Figure 1. Ages (approximate year of life) as at 17 August 2017 for 368,173 cats implanted from 2008 to 2016 and registered on the CAR database. Year of life was approximated as 0 if the cat was aged ≤365 days, 1 if the cat was aged ≤730 days (ie 365+365 days), 2 if the cat was aged ≤1095 days (ie 365+365+365 days, 3 if the cat was aged ≤1461 days (ie 365+365+365+366 days), 4 if the cat was aged ≤1826 days (ie 365+365+365+366+365 days), and so on.

Figure 2. Ages (approximate year of life) as at 17 August 2017 for 848,022 dogs implanted from 2008 to 2016 and registered on the CAR database. Year of life was approximated as 0 if the dog was aged ≤365 days, 1 if the dog was aged ≤730 days (ie 365+365 days), 2 if the dog was aged ≤1095 days (ie 365+365+365 days, 3 if the dog was aged ≤1461 days (ie 365+365+365+366 days), 4 if the dog was aged ≤1826 days (ie 365+365+365+366+365 days), and so on.

Figure 3 and 4 are excellent – display really clear peaks in registration coinciding with reminder emails. But are these both owners that were sent reminders and were not? If this includes both types of owners, perhaps their data could be colour coded to show both populations on the one graph? This may become too difficult to read, so is only a suggestion. Would be good to state what population of owners is included in the figure titles (i.e. received a reminder or did not).

The authors appreciate the suggestion to colour code the data, however the sample size would make colours too difficult to read. Figure title amended to clarify that all owners were included, irrespective of email reminder.

Line 339 – space missing ahead of “In contrast, for animals...”

Amended

Consider a restructure of the results – pairing up the descriptive stats with the odds ratios – e.g. 3.2.4 immediately following 3.1.4, so that the whole question of socio-economics is dealt with in one go. This is only a suggestion, the authors may have good reason to stick with the current structure.

The authors thank you for this suggestion. Our reasoning for structuring the results this way is to make it clear that there were two different sets of analysis on two different study populations.

For consideration for section 4.2: Owners without emails – is updating difficult for them? Perhaps updating online is more straightforward when you have an email address logged with the system. i.e. getting reminders is not the only issue with non-email owners, perhaps these are also non-internet users for which the updating process is more cumbersome? The fact that owner updates were approximately 2 to 3 times more likely for owners who provided an email address but did not receive an email reminder, compared to those who did not provide an email address or receive a reminder suggests this might be the case? Or people who don’t supply an email address move less frequently?

Thank you for your insight. Updating of details is not difficult for owners without an email address listed. Owners without an email address listed can create log-in access via the CAR website, or details can be updated via phone and mail. Thus, owners without an email on file can readily update their details online, and non-internet users have a non-cumbersome alternative available. The following has been added on line 178-182:

·         There are several methods available for owners to update their pets’ microchip details with the CAR database, other than through email reminders. Owners without an email address listed can create log-in access via the CAR website, or details can be updated via phone and mail. Thus, owners without an email on file can readily update their details online, and non-internet users have an alternative available.

For consideration for section 4.3.1: Difference in frequency of updates between locations could be because people move more frequently in one place (e.g. Victoria) as compared to another (e.g. NT)?

This is a valid point, however in this case, it is quite the opposite! The Australian Bureau of Statistics reported that NT had the most transient population of all Australian states and territories.

Line 523 – is this difference between cats and dogs significant? 

Yes (P<0.001). Sentence on line 529 altered to reflect this:

·         Cats (2.5%) had less frequent updates of owner details than dogs (3%; P<0.001).

Reviewer 2 Report

I think this manuscript is quite interesting, but I feel that there are a number of problems needed to be revised. In general, taking into account the aims of the manuscript, some large amount of information is redundant. This makes reading the paper difficult, and discourage readers to go to the end. I would strongly suggest the Authors to keep focused on the major aims (to investigate whether sending email reminders increased the frequency that owners updated their details on the microchip database, and to determine whether there were relationships between the frequency of updates and species (cat or dog), pet age, state/territory, and socioeconomic status of the owner's locality.) and remove the first aim (to describe the characteristics of cats and dogs registered on an Australian national microchip database) even because it is referred to a different experimental period.

However, notwithstanding the limitation of the study highlighted (par. 4.4), the approach is quite innovative and it could be of interest for exploring new strategies aimed to contain the problem of stray pets.  The Authors could better organize the material into a coherent whole.  There is the need to better explain some procedure in the material and methods section to increase clarity. In addition, it would be interesting to explore how factors such pet age or breed or owner socioeconomic index or location differ between the euthanized pets and those returned to the owners. 

Hereby, some adjunctive comments are further provided.

Line 80-100. It would be clearer moving this paragraph before line 62 and following with the statistics and consequences of being microchipped or unmicrochipped, also in relation to the possibility of being reclaimed as a stray pet from the owner for a reunion.

Line 102. The Authors only focused on Anglo Saxon and US countries, not worldwide!

Line 139-141. Please explain better CAR recovery files, it is unclear.

Line 154-155.  How many? “Owners with no email address recorded at the time of their pet's anniversary did not receive any anniversary reminders.”

Line 158. Format: delete “;” after the full stop.

Line 170. Format: live a space in the middle of “mixed-bred.Socioeconomic”

Line 156-202. These sub-sections are quite confusing, in terms of characteristics and timing of the sample chosen. It might be helpful to use a chronogram, highlighting sample size and information gained. Please rewrite them.

The sample size of dogs and cats registered at the CAR at the 1 March 2015, as well as of those belonging to this sample still being registered on 17 August 2017 should be reported.

Line 177-179.  What about the pause from 31 March to 4 July 2016 due to a change in CAR’s information technology systems? Is this included in the chosen period from 1 March 2015 to 17 August 2017 extracted?

Line 185-186. How many?

Line 572.  Spelling “validity”

Line 573-581. I agree this is a confounding factor, that is why it would have been useful to follow even a smaller number of animals, but persisting in the chosen time-frame from 31 March to 4 July 2016. In addition, the assumption that the most deceased animals would have been older animals it is not necessarily true.

I strongly suggest to the Authors to thoroughly revise all the manuscript, resubmitting it afterwards, because I think that their study could be of scientific interest.

Author Response

The authors would like to thank you for taking the time to read our manuscript and provide their suggestions and recommendations. Below are our responses to your comments:

Reviewer 2

I think this manuscript is quite interesting, but I feel that there are a number of problems needed to be revised. In general, taking into account the aims of the manuscript, some large amount of information is redundant. This makes reading the paper difficult, and discourage readers to go to the end. I would strongly suggest the Authors to keep focused on the major aims (to investigate whether sending email reminders increased the frequency that owners updated their details on the microchip database, and to determine whether there were relationships between the frequency of updates and species (cat or dog), pet age, state/territory, and socioeconomic status of the owner's locality.) and remove the first aim (to describe the characteristics of cats and dogs registered on an Australian national microchip database) even because it is referred to a different experimental period.

The authors acknowledge that describing animal characteristics may seem unrelated to the other aims, however, the authors feel that the results gained from this analysis of animal characteristics from such a large animal population is too novel and valuable to readers and shelter medicine to remove from the paper.

However, notwithstanding the limitation of the study highlighted (par. 4.4), the approach is quite innovative and it could be of interest for exploring new strategies aimed to contain the problem of stray pets.  The Authors could better organize the material into a coherent whole.  There is the need to better explain some procedure in the material and methods section to increase clarity. In addition, it would be interesting to explore how factors such pet age or breed or owner socioeconomic index or location differ between the euthanized pets and those returned to the owners. 

We agree that this is a complex dataset and study design which is challenging to explain. Sections 2.1-2.6 have been reworded and reorganised, and Table 2 has been added to help clarify the process.

We also agree that this would be very interesting and valuable information to our paper, but unfortunately this information is not reported in the cited studies.

Hereby, some adjunctive comments are further provided.

Line 80-100. It would be clearer moving this paragraph before line 62 and following with the statistics and consequences of being microchipped or unmicrochipped, also in relation to the possibility of being reclaimed as a stray pet from the owner for a reunion.

Order of introduction paragraphs changed according to suggestion.

Line 102. The Authors only focused on Anglo Saxon and US countries, not worldwide!

Amended line 106 to read:

·         percentages of strays not just in Australia, but in several first world nations.

Line 139-141. Please explain better CAR recovery files, it is unclear.

Line 149-153 has been reworded to read:

·         Recovery records were created when a microchip number was searched for, in the case of lost or stolen pet. If not recorded on CAR (because the number had been previously recorded on one of the other Australian microchip registries), a record on CAR was created, indicating that the registry contained complete, current animal owner information. 

Line 154-155.  How many? “Owners with no email address recorded at the time of their pet's anniversary did not receive any anniversary reminders.”

 These values are presented in Table 5.

Line 158. Format: delete “;” after the full stop.

Amended 

Line 170. Format: live a space in the middle of “mixed-bred.Socioeconomic”

Amended 

Line 156-202. These sub-sections are quite confusing, in terms of characteristics and timing of the sample chosen. It might be helpful to use a chronogram, highlighting sample size and information gained. Please rewrite them.

Sections 2.1-2.6 have been reworded and Table 2 added (line 161):  

Table 2. Explanations and sizes of study populations, according to type of analysis.

Analysis

Study population

Numbers of animals included

Other

Animal characteristics

Cats and dogs with microchip implanted   from 1 January 2008 to 31 December 2016

394,747 cats and 904,909 dogs

Updates of owner details

Cats and dogs first registered with CAR   from 1 March 2015 to 24 February 2016

41,545 cats and 82,942 dogs

Updates from 1 March 2015 to 17 August   2017

The sample size of dogs and cats registered at the CAR at the 1 March 2015, as well as of those belonging to this sample still being registered on 17 August 2017 should be reported.

We have included explanations and sizes of study populations in the newly-created Table 2. We have also added the following explaining that we did not account for registered animals that had subsequently died, etc. in line 193-196:

·         Owners commonly did not record instances of animal deaths, losses or thefts on the database. Accordingly, we could not account for many of these animals in analyses. Rather, we assumed all animals were alive and in the care of the owner (as recorded in the database at 14 August 2017) for at least 540 days from the animal's registration date. 

Line 177-179.  What about the pause from 31 March to 4 July 2016 due to a change in CAR’s information technology systems? Is this included in the chosen period from 1 March 2015 to 17 August 2017 extracted?

The 31 March- 4 July 2016 period refers to email reminders, not updates of owner details. This has been clarified in line 190-191:

·         All updates of owner details from 1 March 2015 to 17 August 2017 were extracted (including updates during the 31 March 2016 to 4 July 2016 period, when email reminders were not sent; Table 2).

Line 185-186. How many?

 (Now line 209-210)

Table 2 has been included to explain study populations and  numbers.

Line 572.  Spelling “validity”

Amended 

Line 573-581. I agree this is a confounding factor, that is why it would have been useful to follow even a smaller number of animals, but persisting in the chosen time-frame from 31 March to 4 July 2016. In addition, the assumption that the most deceased animals would have been older animals it is not necessarily true.

Words regarding older animals removed from section 4.4. 

Reviewer 3 Report

Dear authors,

The paper addresses a very important issue regarding dog and cat abandonment.

The methods are well described and the results are very detailed and properly discussed.

I would only suggest a few modifications to further increase the impact of the paper.

-       References to studies conducted in other countries other than Australia, the US or the UK could be included. In fact, a study paying attention to the influence of microchipping on reclaiming was published in this very journal back in 2015. Animals 2015, 5, 426-441; doi:10.3390/ani5020364

 -       The link between microchipping and reclaiming success could be partially explained by the fact that microchipping is an indicator/predictor of responsible ownership. I believe this should be more clearly stated in the text (e.g. line 65, where you just talk about “factors”).

-       I think it would be interesting to further discuss the impact that increasing updates could have on reclaiming success. You could even make a theoretical projection based on your results and the information provided at the beginning of the paper (L.91-92).

Author Response

The authors would like to thank you for taking the time to read our manuscript and provide their suggestions and recommendations. Below are our responses to your comments:

 Reviewer 3

-       References to studies conducted in other countries other than Australia, the US or the UK could be included. In fact, a study paying attention to the influence of microchipping on reclaiming was published in this very journal back in 2015. Animals 2015, 5, 426-441; doi:10.3390/ani5020364

Thank you, we have used information from this article in line 52-53, 60-61, 82-83:

·         and 68% of cats and dogs admitted to shelters in Spain[9].

·         and in Spanish shelters, 17% of dogs and 4% of cats were reclaimed while 13% of dogs and 23% of cats were euthanized[9].

·         and a study of shelters in Spain showed dogs were three times more likely to be reclaimed if microchipped[9].

 -       The link between microchipping and reclaiming success could be partially explained by the fact that microchipping is an indicator/predictor of responsible ownership. I believe this should be more clearly stated in the text (e.g. line 65, where you just talk about “factors”).

We have inserted this point in line 78-80:

·         Although factors other than presence of a microchip would have contributed to these differences (for example, microchipping could be considered to be an indicator of responsible pet ownership),

-       I think it would be interesting to further discuss the impact that increasing updates could have on reclaiming success. You could even make a theoretical projection based on your results and the information provided at the beginning of the paper (L.91-92).

A new sentence has been added on line 106-108 to further emphasize this relationship:

·         Increasing the proportion of pets with correct contact details is positively correlated with being reclaimed and negatively correlated with euthanasia in shelters.